# Enhancing Adaptive Deep Networks for Image Classification via Uncertainty-aware Decision Fusion

Xu Zhang
xuzhang22@m.fudan.edu.cn
School of Computer Science, Fudan
University, Shanghai, China

Zhipeng Xie
xiezp@fudan.edu.cn
School of Computer Science, Fudan
University, Shanghai, China

Haiyang Yu
hyyu20@fudan.edu.cn
School of Computer Science, Fudan
University, Shanghai, China

Qitong Wang*
qitong.wang@etu.u-paris.fr
Universite Paris Cite, Paris, France

Peng Wang
pengwang5@fudan.edu.cn
School of Computer Science, Fudan
University, Shanghai, China

Wei Wang
weiwang1@fudan.edu.cn
School of Computer Science, Fudan
University, Shanghai, China

## Abstract

Handling varying computational resources is a critical issue in modern AI applications. *Adaptive deep networks*, featuring the dynamic employment of multiple classifier heads among different layers, have been proposed to address classification tasks under varying computing resources. Existing approaches typically utilize the last classifier supported by the available resources for inference, as they believe that the last classifier always performs better across all classes. However, our findings indicate that earlier classifier heads can outperform the last head for certain classes. Based on this observation, we introduce the Collaborative Decision Making (CDM) module, which fuses the multiple classifier heads to enhance the inference performance of *adaptive deep networks*. CDM incorporates an uncertainty-aware fusion method based on evidential deep learning (EDL), that utilizes the *reliability* (uncertainty values) from the first $c$-1 classifiers to improve the $c$-th classifier' accuracy. We also design a balance term that reduces fusion *saturation* and *unfairness* issues caused by EDL constraints to improve the fusion quality of CDM. Finally, a regularized training strategy that uses the last classifier to guide the learning process of early classifiers is proposed to further enhance the CDM module's effect, called the Guided Collaborative Decision Making (GCDM) framework. The experimental evaluation demonstrates the effectiveness of our approaches. Results on ImageNet datasets show CDM and GCDM obtain 0.4% to 2.8% accuracy improvement (under varying computing resources) on popular adaptive networks. The code is available at the link https://github.com/Meteor-Stars/GCDM_AdaptiveNet.

## CCS Concepts

• **Computing methodologies → Computer vision**.

*Corresponding author.

## Keywords

Image classification, adaptive deep networks, deep learning

**ACM Reference Format:**
Xu Zhang, Zhipeng Xie, Haiyang Yu, Qitong Wang, Peng Wang, and Wei Wang. 2024. Enhancing Adaptive Deep Networks for Image Classification via Uncertainty-aware Decision Fusion. In *Proceedings of the 32nd ACM International Conference on Multimedia (MM '24), October 28-November 1, 2024, Melbourne, VIC, Australia.* ACM, New York, NY, USA, 9 pages. https://doi.org/10.1145/3664647.3681368

## 1 Introduction

Deep convolutional neural networks (CNN) include the traditional architecture of ResNet [11] and DenseNet [15] or the light-weight architecture of MobileNet [12] and CondenseNet [14]. They have promoted the development of many fields such as object detection [4]. The above deep networks that contain only one classifier at the end of the network architecture, and they need to work on high computational costs and will become ineffective when computational resources are insufficient.

A popular solution is to transform deep networks into a multi-classifier network, where each classifier works based on different computational resources [13]. As shown in Figure 3, a deep network consists of $c$ blocks, each consisting of different CNNs. Different classifiers are attached to the exits of different blocks. The computational resources required for the $c$-th classifier are the sum of all the preceding $c - 1$ blocks. This enables different classifiers to work under varying computational resources and if it's insufficient to support the $c$-th classifier, we can select a classifier from the first $c - 1$ classifiers that meet the requirements. Deep networks with multi-classifiers can be seen as *Adaptive Deep Networks*. The simplest implementation of it is to add multiple classifiers at different depths in traditional CNN architectures like ResNet [11]. However, traditional architectures may suffer from optimization conflicts between classifiers, leading to poor performance. Encouragingly, a new architecture called MSDNet [13] is proposed and successfully addresses the optimization conflicts among multiple classifiers, subsequently leading to the emergence of a more effective architecture like RANet [28] and their improved versions [1, 20, 29]. Meanwhile, IMTA [19] proposes improved techniques for enhancing adaptive deep networks, which is a gradient equilibrium-based two-stage training algorithm to further enhance the performance of adaptive deep networks like MSDNet and RANet.

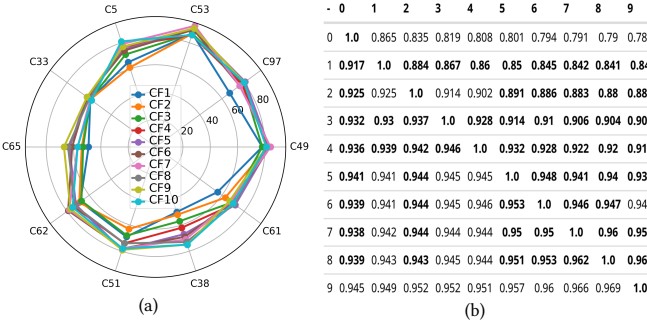

| - | 0 | 1 | 2 | 3 | 4 | 5 | 6 | 7 | 8 | 9 |
|---|---|---|---|---|---|---|---|---|---|---|
| 0 | **1.0** | 0.865 | 0.835 | 0.819 | 0.808 | 0.801 | 0.794 | 0.791 | 0.79 | 0.789 |
| 1 | **0.917** | **1.0** | 0.884 | 0.867 | 0.86 | **0.85** | 0.845 | 0.842 | 0.841 | 0.84 |
| 2 | **0.925** | 0.925 | **1.0** | 0.914 | 0.902 | **0.891** | **0.886** | 0.883 | 0.88 | 0.881 |
| 3 | **0.932** | 0.93 | **0.937** | **1.0** | 0.928 | 0.914 | 0.91 | 0.906 | 0.904 | 0.904 |
| 4 | **0.936** | **0.939** | **0.942** | **0.946** | **1.0** | **0.932** | 0.928 | 0.922 | 0.92 | 0.919 |
| 5 | **0.941** | 0.941 | **0.944** | 0.945 | 0.945 | **1.0** | 0.948 | 0.941 | 0.94 | 0.939 |
| 6 | **0.939** | 0.941 | **0.944** | 0.945 | 0.946 | **0.953** | **1.0** | 0.946 | **0.947** | 0.946 |
| 7 | **0.938** | 0.942 | **0.944** | 0.944 | 0.944 | **0.95** | **0.95** | **1.0** | 0.96 | 0.956 |
| 8 | **0.939** | 0.943 | **0.943** | 0.945 | 0.944 | **0.951** | **0.953** | **0.962** | **1.0** | 0.961 |
| 9 | 0.945 | 0.949 | 0.952 | 0.952 | 0.951 | 0.957 | 0.96 | 0.966 | 0.969 | **1.0** |

(a)  (b)

**Figure 1: Motivation analysis: (a) Accuracy of different classifiers of MSDNet on randomly sampled classes with CIFAR100 dataset. (b) Agreement measurement on 10 classifiers of MSDNet on ImageNet100 with regularized training. A lower value represents higher diversity. The values in bold denote that the agreement value decreases after regularized training.**

However, current *adaptive deep networks* have a common limitation: they assume the last classifier always has the best performance on all classes. Consequently, they only use the last $C$-th classifier that computing resources can support (the previous all $C-1$ classifiers are unused). This raises a question: *when there are enough computational resources for the c-th classifier, it is also sufficient to support all c − 1 classifiers. Can we utilize all the previous c − 1 classifiers to enhance the performance of the c-th classifier?* The answer is affirmative because we find there is good diversity among the first $c$ classifiers in *Adaptive Deep Networks*, and their decisions can complement each other.

As shown in Figure 1(a), we visualize the accuracy of randomly sampled classes (C5, C53,...) on different classifiers. CF1 and CF10 represent classifiers attached to the 1-th and 10-th block, and they are allocated with the minimal and maximum computational resources. We can observe that the overall performance of CF10 is the best among all classifiers because it utilizes the maximum resources to extract the highest-quality features for classification. However, the accuracy of CF10 in some classes (e.g., C53 and C65) is not the best and is even worse than other classifiers. This suggests that different classifiers have their own advantages (more qualitative analysis refers to the full version paper shown in our code link), and $c-1$ classifiers can provide better decisions for the $c$-th classifier to enhance its performance.

Based on the above observations, this paper proposes CDM, called Collaborative Decision Making, which fuses the decision information of all $c-1$ classifiers to enhance the performance of the $c$-th classifier. CDM works only during the inference stage with no extra model parameters and without obviously increasing inference time. For the design of CDM, we propose a classifier fusion method based on evidential deep learning (EDL) [25] framework and uncertainty. Specifically, classifiers with higher uncertainty are considered to have less *reliability*, and vice versa. To this end, we first propose an uncertainty-aware attention mechanism-based fusion method to weight and integrate the decision information from different classifiers based on their *reliability*. However, we find that the prior design in the EDL framework may bring *fusion saturation* and *fusion unfairness* issues for the uncertainty-aware fusion, which harms its fusion quality. Hence, a balance term to

slow down the changing trend of fusion values is further introduced to alleviate the *fusion saturation* and *fusion unfairness* issues, enhancing the fusion quality of CDM.

Finally, we make efforts to enhance the performance of the CDM module. The fusion performance of multiple classifiers depends on the accuracy and diversity of them [2, 3, 23]. For accuracy, as described in Figure 1(a), the overall performance on all classes of the last classifier is better than the early classifiers. Hence, we can increase the accuracy of early classifiers by exerting regularization between the last classifier and early ones (called *regularized training*). To this end, we propose the Guided Collaborative Decision Making (GCDM) framework to use the last classifier to guide the learning process of early classifiers. For diversity, intuitively, *regularized training* may decrease the diversity of early classifiers. However, we observe that *regularized training* doesn't obviously harm and can even improve the diversity among classifiers in experiments. Specifically, we calculate the agreement table [27]) of 10 classifiers of MSDNet after *regularized training* on the ImageNet100 dataset, as shown in Figure 1(b). Therefore, we can enhance the performance of CDM by *regularized training*, i.e., increasing the accuracy of early classifiers, not obviously harming and even improving their diversity. Our contributions are as follows:

(1) *A Collaborative Decision Making (CDM) idea* is proposed to improve the performance of popular *adaptive deep networks*.

(2) *An uncertainty-aware fusion method* is proposed to realize CDM, which weights and integrates the decision information from different classifiers based on their *reliability* (uncertainty values).

(3) *A Guided Collaborative Decision Making (GCDM) framework* is further proposed to enhance CDM, which uses regularized training to increase the accuracy of early classifiers and not obviously harm their diversity.

(4) *Empirical study on the large scale ImageNet1000*, ImageNet100, Cifar100, and Cifar10 datasets shows the good performance of CDM and GCDM, e.g., our method can improve the accuracy of SOTA MSDNet, RANet, and IMTA by approximately 0.8% to 2.8% under various computing resource constraints on the ImageNet datasets.

## 2 Related Work

***Adaptive deep networks.*** Works can be divided into two categories: focusing on designing effective architectures and training strategies. For the former one, MSDNet [13] creates an innovative multi-scale convolutional network featuring multi-classifiers with different computational budgets. These classifiers can be dynamically chosen during the inference stage. RANet [28] proposes a resolution adaptive network by designing an architecture that can feed features with a suitable resolution for different samples. Then, [1] uses concatenation and strided convolutions to further improve MSDNet. [20] proposes an adaptive router to predict the difficulty scores of the images and achieve automatic classification. [29] regards *adaptive deep networks* as an additive model, and train it in a boosting manner to address the distribution mismatch problem in the train-test stages. For designing effective training strategies, IMTA [19] proposes improved techniques, such as the gradient equilibrium and forward-backward knowledge transfer based two-stage training algorithms (introducing extra model parameters) for improving the performance of *adaptive deep networks*. However,

there is a limitation for current methods: they all assume the last classifier always has the best performance and multi-classifiers work independently during the testing stage. In other words, when computational resources are sufficient to support the $c$-th classifier, all $c-1$ classifiers can also be used to improve the performance. However, they ignore the decision information of the $c-1$ classifiers, resulting in computational resource waste and performance limitations. This paper proposes one-stage (end-to-end training) methods to address this limitation, making full use of computational resources and enhancing the accuracy of each classifier.

**Evidential Deep Learning (EDL).** Traditional softmax-based neural networks for single-point estimation of class probability distributions cannot effectively estimate classification uncertainty and are prone to overconfidence in wrong predictions [8]. In contrast, EDL targets knowing "what they don't know" based on a prior belief [21, 26]. Through the Dempster Shafer Theory of Evidence (DST) [6] and Subjective Logic [16], EDL realizes uncertainty estimation in a single forward pass by collecting evidence for each category and modeling the distribution of class probabilities. In recent years, EDL has successfully been adopted in various tasks, including out-of-distribution detection [7, 30], multiview classification [9, 10] and domain adaptation learning [5]. However, to our knowledge, EDL hasn't been explored in the field of *adaptive deep networks*. This paper uses evidential uncertainty to quantify the *reliability* of different classifiers and further develops an uncertainty-aware attention mechanism to fuse the decision information from different classifiers with an emphasis on their *reliability*.

## 3 Methods

Our method is shown in Figure 3. Unlike traditional *adaptive deep networks*, our proposed approach CDM differs in that during the inference stage, the $c$-th classifier and all $c-1$ classifiers are not independent, thereby making full use of computational resources. Specifically, we enhance the performance of the $c$-th classifier through the proposed CDM module and GCDM framework. In CDM, an uncertainty-aware attention mechanism is designed to weight and fuse the decision information from different classifiers based on their *reliability* (uncertainty values). Further, through regularized training, GCDM enhances the performance of CDM by increasing the accuracy of early classifiers and not obviously harming or even improving their diversity. We will proceed to introduce the details of CDM and GCDM.

### 3.1 Problem Definition

The *adaptive deep network* can be seen as a network with $C$ classifiers, where these classifiers are attached at varying depths of the network. Given the input image $x$ and corresponding true class label $y$, the output of the $c$-th classifier ($1 \leq c \leq C$) is:

$$p^c = f_c(x; \theta_c) = [p_1^c, \cdots, p_K^c] \in \mathbb{R}^K \qquad (1)$$

where $K$ is the number of class labels and $\theta_c$ denotes the model parameters of the $c$-th classifier and each value $p_k^c$ is the logit of the $k$-th class on $c$-th classifier.

The *adaptive deep network* can realize computationally efficient inference in two forms: *anytime prediction* (Figure 2(a)) and *budgeted batch prediction (Figure 2(b))*. As shown in Figure 2, consider

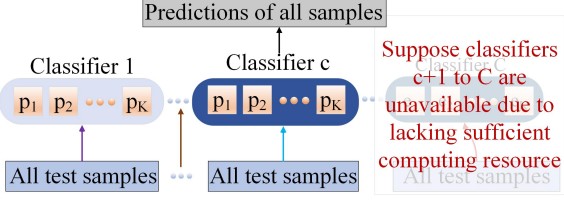

**(a)** Traditional *Anytime prediction* setting ($c$-1 classifiers are unused)

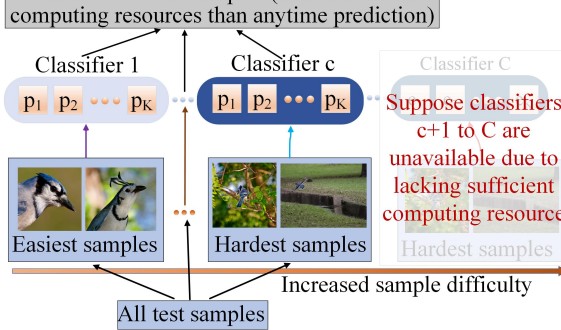

**(b)** Traditional *Budgeted batch prediction* setting  ($c$-1 classifiers are unused)

**Figure 2: Comparison between *anytime prediciton* and *budgeted batch prediction* settings.**

the computational resource is only sufficient for the $c$-th classifier. In *anytime prediction*, all test samples will be classified by the $c$-th classifier. In contrast, *budgeted batch prediction* dynamically selects the suitable classifier for different test samples based on their difficulty. Concretely, early classifiers with fewer computational resource costs are used for the classification of easy images, while classifiers with higher computational demands are used for the classification of harder images. To measure the difficulty of images, we follow [13, 19, 28] to use the classifier confidence of validation set. Hence, the frequency of using classifiers with higher computational resource will be reduced, thereby achieving a further reduction in overall computational resource consumption. We recommend to see [13, 28] for more details about *budgeted batch prediction*.

### 3.2 Collaborative Decision Making (CDM)

CDM is proposed to enhance the performance of $c$-th classifier by reusing the $c-1$ available classifiers under limited computational resources where $1 < c \leq C$ and $C$ is the total number of classifiers. CDM first quantifies the uncertainty (*reliability*) of different classifiers and uses the uncertainty-aware attention mechanism for collaborative decision-making fusion. We will further discuss them.

*3.2.1 Quantify the Uncertainty of Classifiers (QUC).* Evidential uncertainty is derived from the Dempster–Shafer Theory of Evidence (DST) and is further developed by Subjective Logic (SL) [17] based on Dirichlet distribution. SL defines a theoretical framework for obtaining the belief masses of different classes and the uncertainty measurement of the samples based on the *evidence* collected from them. For the $c$-th classifier, $K + 1$ mass values are all non-negative and their sum is one:

$$u^c + \sum_{k=1}^{K} b_k^c = 1 \qquad (2)$$

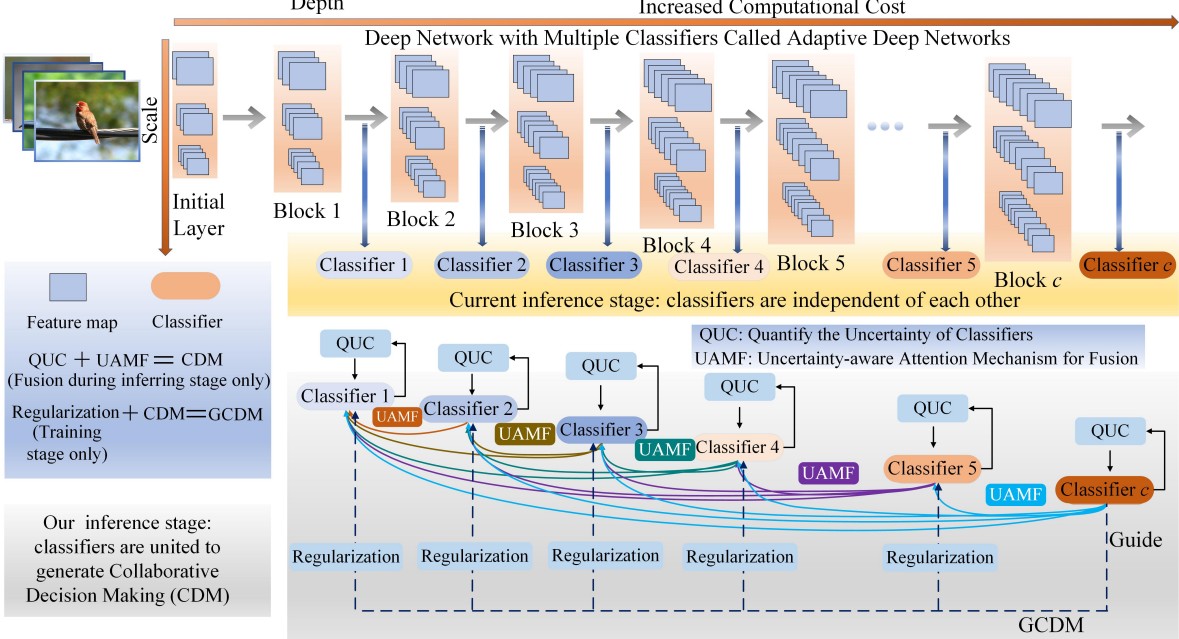

**Figure 3: Overview of our methods for *adaptive deep network*. Our proposed CDM fusion is suitable for both *anytime prediction* and *budgeted batch prediction* settings as shown in Figure 2.**

where $b_k^c$ represents belief mass (the probability of the $k$-th class on the $c$-th classifier) and $u^c$ is uncertainty. The Dirichlet parameters $\alpha^c$, evidence $e^c$, belief mass $b_k^c$ and uncertainty $u^c$ are defined as:

$$\alpha_k^c = e_k^c + 1 = SoftPlus(p_k^c) + 1, b_k^c = \frac{e_k^c}{S^c}, u^c = \frac{K}{S^c} \quad (3)$$

where $p_k^c$ is the output of these classifiers, as described in Eq. 1 and $S^c = \sum_{k=1}^{K}(e_k^c + 1)$ is Dirichlet strength. $SoftPlus(\cdot)$ is a smoothed ReLU activation function.

We can quantify the uncertainty $u^c$ by optimizing the Dirichlet distributions [30] of different classifiers:

$$\mathcal{L}_u = \sum_{c=1}^{C} \sum ((b^c - y)^2 + Var(f_c(x; \theta_c))) \quad (4)$$

where $Var(f_c(x; \theta_c))$ denotes the variance of the Dirichlet distribution.

*3.2.2 Uncertainty-aware Attention Mechanism for Fusion (UAMF).* The obtained uncertainty $u^c$ can measure the *reliability* of $c$-th classifier. Then, we design an uncertainty-aware attention mechanism to weight and fuse the decisions from different classifiers based on their *reliability* (uncertainty values). The key design is to focus more on classifiers with higher reliability (lower uncertainty) during decision fusion. For simplicity, we take the fusion of the two classifiers for instance and the proposed uncertainty-aware fusion process is formulated by:

$$\hat{u} = u^c u^{c+1} \quad (5)$$

$$\hat{b}_k = b_k^c b_k^{c+1} + b_k^c (1 - u^c) + b_k^{c+1}(1 - u^{c+1}) \quad (6)$$

$$\hat{S}^c = K/\hat{u}^c, \hat{e}_k^c = \hat{S}^c \cdot \hat{b}_k \quad (7)$$

We first fuse uncertainty $u$ and belief mass $b$ through element-wise multiplication. Such form ($u^c u^{c+1}$ and $b_k^c b_k^{c+1}$) has the advantage of enhancing compatible information while suppressing conflicting portions, helping gather information where the two

classifiers reach a consensus. Then, we use $1 - u^c$ and $1 - u^{c+1}$ as attention weights (also called *attention term*) to weight the decision information from $b_k^c$ and $b_k^{c+1}$. When the uncertainty value is high, we reduce the contribution of this classifier to the fusion because it shows low *reliability*, and vice versa. Further, the fused evidence $\hat{e}_k^c$ is used for final classification.

However, the accumulative multiplication form in Eq. 5 and 6 will cause two issues. First, in Eq. 2, it holds that $u^c < 1$ and $b_k^c < 1$. If one of them approaches 1 prematurely (e.g., it approaches 1 only in the first 3 classifiers after fusion), the fusion value will change very slowly, which means the fusion between classifiers will be invalid prematurely and the $c$-th classifier cannot obtain useful knowledge from the $(c-1)$ classifiers. We call this issue as *fusion saturation*, as shown in Figure 6 (a). If it appears earlier, the accuracy of the $c$-th classifier will decrease after fusion with $(c-1)$ classifiers and even be worse than a single $c$-th classifier without fusion.

Second, before the appearance of the *fusion saturation*, the fusion values will change sharply due to the accumulative multiplication form of fusion. In Eq. 5- 7, taking $b_k^c$ for instance here, it will increase sharply after fusion if the classifier to be fused shows the high confidence of the $\hat{k}$-th class (presents high $b_{\hat{k}}^{c-1}$). Moreover, $b_0^c, b_1^c, ..., b_k^c$ where $k! = \hat{k}$ will decrease sharply after fusion at the same time. we call this issue as *fusion unfairness*. Hence, if the $(c-1)$-th classifier presents the fake high confidence towards the $\hat{k}$-th class during fusion, the current sample almost cannot be classified correctly in the latter fusion. One reason is that the belief mass of $\hat{k}$ class is too high to be adjusted. Another reason is the *fusion saturation* issue mentioned above, which leads to invalid fusion because the fusion value changes very slowly. Hence, *fusion saturation* and *fusion unfairness* issues may result in generating an overconfident fusion result, leading to wrong classification.

To relieve the *fusion saturation* and *fusion unfairness* issues, we introduce the *balance term* into Eq. 6 and Eq. 5. Specifically, we introduce weighting and sum operation into the fusion process for the purpose of slowing down the changing trend of fusion parameters uncertainty and belief mass:

$$\widetilde{u} = \zeta + u^c u^{c+1} \tag{8}$$

$$\widetilde{b}_k = (\gamma + b_k^c b_k^{c+1}) \cdot 0.5 + b_k^c(1 - u^c) + b_k^{c+1}(1 - u^{c+1}) \tag{9}$$

$$\gamma = (b_k^c + b_k^{c+1}) \cdot 0.5, \ \zeta = u^c + u^{c+1} \tag{10}$$

where newly added Eq. 10, coefficient 0.5 in Eq. 5 and Eq. 6 is the *balance term* to improve the fusion quality. For obtaining the fused decision of $c$-th classifier, all $c - 1$ classifiers will be sent to the CDM for fusion based on uncertainty-aware attention mechanism, where $c \geqslant 2$. Finally, the fused evidence $\widetilde{e}_k^c$ based on *balance term* can be obtained through Eq. 7 for final classification. The algorithm pseudocode of uncertainty-aware fusion is shown in Algorithm 1.

## 3.3  Guided Collaborative Decision Making

The fusion performance of multiple classifiers depends on the accuracy and diversity of them [2, 3, 23]. Hence, if we can improve the accuracy of early classifiers and their diversity, the performance of CDM can be enhanced. Considering that the overall performance of the last classifier across all classes is better than early classifiers, we exert regularization between the last classifier and others (called *regularized training*), using the last one to guide the learning process of early classifiers. Specifically, the Jensen-Shannon divergence [22] is used to pull close the distribution of the last classifier and early ones. We name the CDM based on *regularized training* as Guided Collaborative Decision Making (GCDM). To make the distribution of the last classifier and the early one more distinguishable, we use temperature-scaled distribution[8] of the classifier logits $f_c(x, \theta_c; \tau)$ instead of original distribution $f_c(x; \theta_c)$:

$$JS(x; c, \tau) = JS\left(f_C(x, \theta_C; \tau) \| f_c(x, \theta_c; \tau)\right) \tag{11}$$

$$f_C(x, \theta_C; \tau) = SoftPlus(p_C)/\tau = e_C/\tau \tag{12}$$

where $c \neq C$ and $JS(\cdot \| \cdot)$ denotes the Jensen-Shannon divergence. We minimize the Jensen-Shannon divergence between the last classifier ($f_C(x, \theta_C; \tau)$) and early classifiers.

The final loss function to optimize GCDM is:

$$\mathcal{L}_u = \mathcal{L}_{JS} + \sum_{c=1}^{C} \sum \left((b^c - y)^2 + Var(f_c(x; \theta_c))\right) \tag{13}$$

$$\mathcal{L}_{JS} = \sum_{c=1}^{C-1} \left(JS(x; c, \tau_1) + JS(x; c, \tau_2)\right)/2 \tag{14}$$

where $C$ denotes the number of classifiers in *adaptive deep networks*. $\tau_1$ and $\tau_2$ are designed to alleviate the performance instability issue in training due to the distance being too long or too short between early classifiers and the last classifier. Hence, this is a Stable Training Strategy (STS) and we calculate the regularization loss twice based on $\tau_1$ and $\tau_2$ instead of using only one $\tau$.

Note that model optimization during training doesn't involve the fusion of classifiers (CDM). CDM only works during the inference stage. The overview of the proposed method is shown in Figure 3.

---

**Algorithm 1:** Algorithm for our uncertainty-aware fusion.

**Input:** $C$ classifier outputs $P = \{p^1, ..., p^c, ..., p^C\}$.
**Output:** $C - 1$ fused decisions E=$\{\widetilde{e}^2, ..., \widetilde{e}^c, ..., \widetilde{e}^C\}$

1  **for** $c = 1 \ to \ C - 1$ **do**
2     **if** *c=1* **then**
3        obtain $(b^1, b^2)$, $(u^1, u^2)$ through Eq. 3
4        obtain fused $\widetilde{u}^2$ based on $(u^1, u^2)$ and Eq. 8
5        obtain fused $\widetilde{b}^2$ based on $(b^1, b^2)$ and Eq. 9
6        obtain $\widetilde{e}^2$ based on $(\widetilde{u}^2, \widetilde{b}^2)$ and Eq. 7
7     **else**
8        obtain $b^{(c+1)}$, $u^{(c+1)}$ through Eq. 3
9        obtain fused $\widetilde{u}^{c+1}$ based on $(\widetilde{u}^c, u^{(c+1)})$ and Eq. 8
10       obtain fused $\widetilde{b}^{c+1}$ based on $(\widetilde{b}^c, b^{(c+1)})$ and Eq. 9
11       obtain $\widetilde{e}^{(c+1)}$ based on $(\widetilde{u}^{c+1}, \widetilde{b}^{c+1})$ and Eq. 7
12    **end**
13 **end**
14 Use $\widetilde{e}^c$ to obtain fused $c$-th classifier accuracy ($c \geq 2$)

---

## 4  Experiments

### 4.1  Experimental Setup

**Datasets.** We use large-scale ImageNet1000, ImageNet100, CIFAR10, and CIFAR100 datasets in experiments. Following [13, 19, 28], all datasets are divided into training, validation, and testing sets. The batch size for the ImageNet100 dataset is fixed at 64 for all methods. Other settings about datasets are as same as in previous work based on their public source codes.

**Baselines.** For *adaptive deep networks*, we use advanced MS-DNet [13], RANet [28] and IMTA [19] to create strong baselines. IMTA is an advanced improved technique for *adaptive deep networks*. For MSDNet, according to the number of blocks between classifiers, there exist two different structures: "E" structure MSDNet$^E$ (the number of blocks between classifiers is equidistant) and "LG" structure MSDNet$^{LG}$ (the number of blocks between classifiers is linearly growing). For RANet, similarly, if the number of layers in each *ConvBlock* is the same or linearly growing, called RANet$^E$ or RANet$^{LG}$ respectively. Detailed model structures refer to the appendix. We don't compare with ensembling multiple independent networks because it performs worse than MSDNet and RANet, as proved in their papers. **For fusion methods baselines,** we select averaging fusion, weighted averaging fusion, voting fusion, neural network fusion, and multiview fusion [10] based on EDL.

**Hyperparameters.** CDM doesn't involve hyperparameters. For GCDM, the hyperparameters $\tau_1$ and $\tau_2$ in Eq. 14 are set to 0.5 and 1, respectively. Overall, our method involves only a few hyperparameters. For hyperparameters of baselines, we directly use the setting in their public source codes. **To ensure a fair comparison, our method is used with the same hyperparameters when combined with other methods. The only difference is whether applying our CDM or GCDM to them**. All experiments in this study are conducted on NVIDIA GeForce RTX 3090 GPU based on PyTorch. More experimental details, results, and discussion can be seen in the full version paper (can be found in our code link).

Xu Zhang, Zhipeng Xie, Haiyang Yu, Qitong Wang, Peng Wang, and Wei Wang

**Table 1: Results of combining our GCDM with MSDNet (*MSD*) and RANet (*RAN*) on the anytime prediction setting. CF*c* represents the *c*-th classifier. *Our* means the corresponding adaptive network equipped with our proposed GCDM. The network of *MSD* and *RAN* both adopt a "*LG*" structure. "-" indicates that there is no *c*-th classifier in the current network. "FLOPs" denotes Floating Point Operations Per Second and we show the average FLOPs of *MSD* and *RAN* for each classifier.**

| Methods/ Classifier(FLOPs) | CIFAR10 | | | | CIFAR100 | | | | ImageNet100 | | | | ImageNet1000 | | | |
|---|---|---|---|---|---|---|---|---|---|---|---|---|---|---|---|---|
| | *MSD* | *Our* | *RAN* | *Our* | *MSD* | *Our* | *RAN* | *Our* | *MSD* | *Our* | *RAN* | *Our* | *MSD* | *Our* | *RAN* | *Our* |
| CF1 ($0.15\times10^9$) | 87.77 | **88.32** | 89.73 | **90.4** | 60.21 | **60.64** | 65.18 | **65.18** | 64.41 | **66.05** | 66.33 | **67.45** | 54.49 | **55.184** | 56.468 | **56.945** |
| CF2 ($0.26\times10^9$) | 90.25 | **90.88** | 91.13 | **91.93** | 63.33 | **66.61** | 68.57 | **71.14** | 68.44 | **70.07** | 69.25 | **70.57** | 61.11 | **61.396** | 63.274 | **63.558** |
| CF3 ($0.39\times10^9$) | 91.7 | **92.01** | 91.85 | **92.89** | 67.82 | **70.36** | 69.27 | **73.26** | 72.26 | **73.53** | 70.67 | **73.06** | 66.85 | **66.986** | 65.996 | **66.893** |
| CF4 ($0.56\times10^9$) | 92.88 | **92.99** | 92.31 | **93.0** | 69.63 | **73.19** | 70.6 | **74.43** | 74.51 | **76.46** | 71.48 | **74.31** | 70.382 | **71.048** | 68.018 | **69.162** |
| CF5 ($0.75\times10^9$) | 93.31 | **93.75** | 93.02 | **93.28** | 72.94 | **75.06** | 73.6 | **76.0** | 76.3 | **78.25** | 72.68 | **75.21** | 72.324 | **73.413** | 68.576 | **70.04** |
| CF6 ($0.88\times10^9$) | 93.58 | **93.84** | 93.02 | **93.43** | 74.17 | **76.14** | 74.11 | **76.67** | 76.56 | **78.78** | 73.3 | **75.5** | 73.018 | **74.288** | 69.614 | **70.756** |
| CF7 ($0.94\times10^9$) | 93.69 | **93.92** | 93.68 | **93.6** | 75.28 | **76.81** | 75.02 | **77.24** | - | - | 75.86 | **77.01** | - | - | 72.492 | **73.069** |
| CF8 ($1.1\times10^9$) | - | - | 93.61 | **93.65** | - | - | 75.48 | **77.39** | - | - | 76.14 | **77.52** | - | - | 73.006 | **73.722** |
| Increased Accuracy | 0.316 (Avg.) 0.630 (Max) | | 0.479 (Avg.) 1.04 (Max) | | 1.93 (Avg.) 3.56 (Max) | | 2.44 (Avg.) 3.99 (Max) | | 1.33 (Avg.) 2.22 (Max) | | 1.86 (Avg.) 2.83 (Max) | | 0.52 (Avg.) 1.27 (Max) | | 0.84 (Avg.) 1.46 (Max) | |

## 4.2 Accuracy under Anytime Prediction and Budgeted Batch Prediction Settings

We combine our GCDM with popular *adaptive deep networks* on various datasets and two settings. **The results of *anytime prediction* setting** are shown in Table 1. Our method brings a stable performance improvement to the current popular networks, whether on large-scale datasets ImageNet1000, or other datasets (CIFAR10, CAIFR100, and ImageNet100). For RANet, the average improvement in accuracy on CIFAR10, CIFAR100, ImageNet100, and ImageNet1000 is 0.479%, 2.44%, 1.86%, and 0.84%, respectively. The maximum accuracy improvement is 1.04%, 3.99%, 2.83%, and 1.46%, respectively. **Moreover, the results of *budgeted batch prediction* setting** in Figure 4 (ImageNet100 and ImageNet) and Figure 5 (CIFAR10 and CIFAR100) also prove that our GCDM consistently improves the classification accuracy of popular *adaptive deep networks* such as MSDNet and RANet by a large margin under the same computational resources (measured by FLOPs). The consistent improvements in the above two settings demonstrate the effectiveness of GCDM. Besides, although the experiments were conducted on the "*LG*" network structures, we still observed consistent performance improvements with our GCDM on the "*E*" structures, which can be found in the full version paper in our code link.

## 4.3 Ablation Study

**Ablation of each component.** We conduct the ablation study on MSDNet with CIFAR100 and ImageNet100 datasets to explore the effectiveness of the proposed CDM and GCDM. The results are shown in Table 2. The proposed CDM significantly improves the performance of the baseline method MSDNet. Moreover, most classifiers among MSDNet equipped with regularized training ($G^+$) perform better than the original one ($0^+$), denoting $G^+$ improves the accuracy of early classifiers. Furthermore, because regularized training doesn't obviously reduce the diversity of the early classifiers, it can further enhance the fusion performance of CDM. Consequently, we

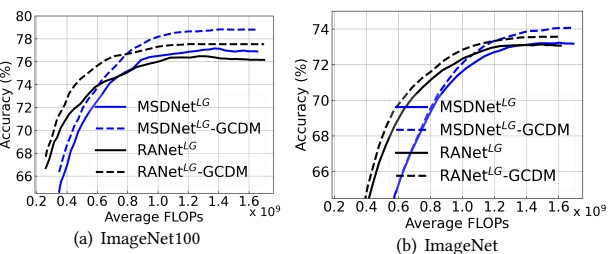

**Figure 4: Accuracy (top-1) of *budgeted batch prediction* on ImageNet100 and ImageNet1000. With the same computational resources, existing methods equipped with the proposed GCDM can achieve better performance.**

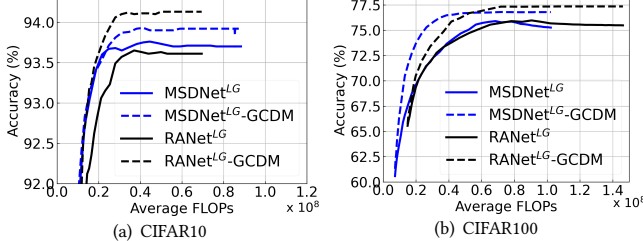

**Figure 5: Accuracy (top-1) of *budgeted batch prediction* on CIFAR10 and CIFAR100.**

observe that MSDNet equipped with GCDM ($G^+ + CDM^+$) obtains better accuracy than $G^+$ and $CDM^+$. The above results validate the effectiveness of the design of CDM and GCDM.

**Diversity of early classifiers after regularization (Eq. 14).** We have proved that regularized training can raise the accuracy of early classifiers. Whether it can improve CDM critically depends on whether it would harm the diversity of early classifiers. Hence, we calculate the diversity metrics of all classifiers of MSDNet$^E$ on CIFAR100 and ImageNet100. The results shown in Table 3 demonstrate that regularized training even can slightly increase the diversity

**Table 2: Ablation study on MSDNet$^E$ with 10 classifiers. $0^+$ denotes the original MSDNet with no fusion for $c$-th classifier. $G^+$ denotes using regularized training. $CDM^+$ denotes using our uncertainty for fusing $c$-th classifier. $G^+$+$CDM^+$ denotes conducting our fusion for a classifier based on regularized training. The best results are in bold while the second best are underlined.**

| Dataset | CF1 | CF2 | CF3 | CF4 | CF5 | CF6 | CF7 | CF8 | CF9 | CF10 |
|---|---|---|---|---|---|---|---|---|---|---|
| ImageNet100 ($0^+$) | 64.36 | 70.39 | 72.99 | 75.93 | 77.0 | 77.09 | 77.37 | 77.65 | 78.01 | 77.93 |
| ImageNet100 ($CDM^+$) | 64.36 | 70.1 | 73.61 | 76.35 | 77.51 | 78.61 | 78.69 | 78.83 | 79.16 | 78.84 |
| ImageNet100 ($G^+$) | 67.32 | 71.32 | 73.43 | 75.78 | 77.38 | 77.84 | 78.2 | 78.16 | 78.55 | 78.69 |
| ImageNet100 ($G^+$+$CDM^+$) | **67.32** | **71.98** | **74.62** | **76.56** | **78.25** | **79.2** | **79.22** | **79.64** | **79.49** | **79.74** |
| Cifar100 ($0^+$) | **63.71** | 66.57 | 68.12 | 70.42 | 72.0 | 72.59 | 73.04 | 74.03 | 74.27 | 74.8 |
| Cifar100 ($CDM^+$) | 63.71 | 68.73 | 71.07 | 73.12 | 74.24 | 75.13 | 75.51 | 76.11 | 76.22 | 76.38 |
| Cifar100 ($G^+$) | 63.1 | 66.39 | 68.53 | 70.44 | 72.12 | 73.37 | 73.46 | 74.38 | 74.39 | 74.74 |
| Cifar100 ($G^+$+$CDM^+$) | 63.1 | **68.56** | **71.55** | **73.41** | **74.65** | **75.48** | **76.05** | **76.44** | **76.79** | **77.03** |

**Table 3: Diversity metrics of correlation coefficient (Cor.), Q-statistic(Q-sta.), Kohavi-Wolpert variance (Var.) [18] and agreement value(Agr.) [27] on MSDNet$^E$ (equipped with 10 classifiers). $G^+$ denotes regularized training; $0^+$ is the opposite. ↓ means lower is better and vice versa.**

| Dataset | Cor.(↓) | Q-sta.(↓) | Var.(↑) | Agr.(↓) |
|---|---|---|---|---|
| Cifar100 ($0^+$) | 0.694 | 0.927 | 0.071 | 0.9006 |
| Cifar100 ($G^+$) | **0.688** | **0.923** | **0.072** | **0.8988** |
| Mi-ImageNet ($0^+$) | 0.737 | 0.957 | 0.0555 | 0.9249 |
| Mi-ImageNet ($G^+$) | **0.733** | **0.955** | **0.0561** | **0.9243** |

among multi-classifiers, ensuring performance improvement for CDM. Figure 1(b) also can prove this point.

**Stable Training Strategy (STS) in loss function Eq. 14.** Results in Figure 8(b) reveal an interesting observation. When using either $\tau = 1$ or $\tau = 0.5$, individually, RANet$^E$ cannot consistently achieve better performance. For example, RANet$^E$ trained with $\tau = 1$ performs better for the first four classifiers while it trained with $\tau = 0.5$ performs better for the latter four classifiers. This shows performance instability issues. However, after introducing STS (using both $\tau = 1$ and $\tau = 0.5$ for weighted JS loss) during training, RANet$^E$ exhibits better performance in most classifiers. This suggests that the introduced STS is effective in relieving the performance instability issue.

## 4.4 Effectiveness of Uncertainty-aware Fusion

**Table 4: Comparison with other fusion methods. The best results are in bold and the second best is underlined.**

| Method | CF5 | CF6 | CF7 | CF8 |
|---|---|---|---|---|
| RANet (without fusion) | 69.072 | 70.016 | 72.55 | 72.95 |
| RANet-average | 68.37 | 69.274 | 70.792 | 71.814 |
| RANet-average$_{weighted}$ | 69.174 | 69.87 | 71.536 | 72.528 |
| RANet-vote | 68.052 | 69.088 | 70.294 | 71.32 |
| RANet-NN$_{weighted}$ | 68.498 | 69.418 | 70.95 | 71.84 |
| RANet-multiview (EDL) | 68.674 | 69.594 | 71.09 | 72.26 |
| RANet-CDM (no balance term) | 68.76 | 69.728 | 71.748 | 72.894 |
| RANet-CDM (no attention term) | 64.824 | 68.294 | 67.268 | 8.82 |
| RANet-CDM (our fusion) | **70.04** | **70.756** | **73.069** | **73.722** |

**Compared with different decision fusion methods.** To evaluate the effectiveness of the proposed uncertainty-aware fusion,

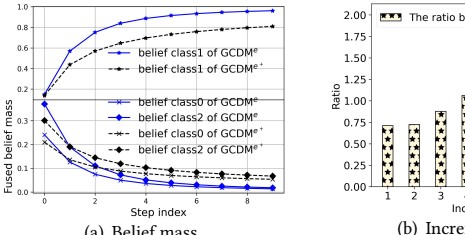

(a) Belief mass          (b) Increment ratio

**Figure 6: Analysis for the value changing trend between our fusion with *balance term* and the original one.**

we compare our method with different decision fusion methods on large-scale ImageNet1000 dataset, including traditional average, weighted average, voting, neural network fusion method, as well as multi-view fusion method [9] based on evidential learning. For the weighted average fusion, we normalize the accuracy of the classifiers on the validation set to obtain the weights for each classifier. For Neural Network (NN) weighted fusion, we allocate an MLP for $c$-th ($c \geq 2$) classifier for fine-tuning based on the well-trained *adaptive deep networks*. The comparison results are shown in Table 4. We observe that traditional and multi-view fusion methods are even poorer than the original RANet while our fusion is better than all other methods. This is because traditional fusion methods don't take into account uncertainty, and multi-view fusion doesn't consider the issues of *fusion saturation* and *fusion unfairness*.

**Effectiveness of designed *balance term* and *attention term*.** We conduct the fusion experiment on ImageNet1000 for CDM with *balance term* (abbreviation is GCDM$^{e+}$) and without *balance term* (abbreviation is GCDM$^e$). We visualize the changing trend of the believe mass $\hat{b}_k$ under different times of fusion, as shown in Figure 6(a). We find that GCDM$^{e+}$ successfully slows down the changing trend of the fusion process, relieving the *fusion saturation* and *fusion unfairness* issues. In other words, GCDM$^{e+}$ won't lead to prematurely *fusion saturation*. The high belief mass of a certain class (e.g., 1) won't lead to a sharp decrease in the belief mass of other classes (e.g., 0 and 2), which means that the *fusion unfairness* issue is relieved. We also record the belief mass increment between $c$-th and $(c-1)$-th fusion for both GCDM$^{e+}$ and GCDM$^e$. Finally, we calculate the increment ratio between GCDM$^{e+}$ and GCDM$^e$, as shown in Figure 6(b). We find that the belief mass increment ratio is larger than GCDM$^e$ with the increase of fusion times, which ensures the effectiveness in the following fusion operations. Hence, GCDM$^{e+}$ can obtain better performance than GCDM$^e$, proving the

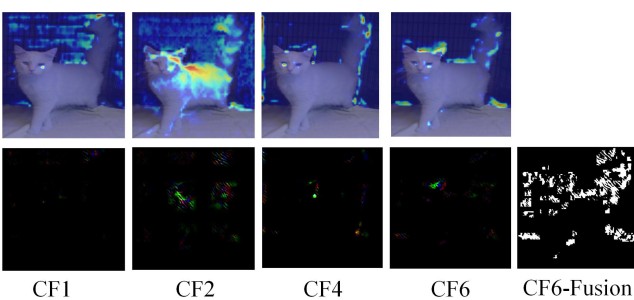

CF1          CF2          CF4          CF6     CF6-Fusion

**Figure 7: Interested regions visualization based on classifiers in MSDNet$^E$.**

effectiveness of our designed *balance term*. As shown in Table 4, we also can observe that the performance of RANet-CDM decreases after removing the balance term, further proving the effectiveness of designed *balance term*. Moreover, performance sharply declines after further removing the *attention term*, especially for the last classifier CF8. This indicates that the basic *attention term* is crucial in the uncertainty-aware fusion.

**Interested regions visualization of different blocks in MS-DNet.** As shown in Figure 7, based on the cat and dog dataset, we visualize the interested regions of different blocks in the well-trained MSDNet$^E$. The top row is visualized on Grad-CAM and the bottom row is on Guided Grad-CAM[24] for pixel-level visualization. We can find that different classifiers capture different regions. Even more interesting is that we use the fusion result of the total 6 classifiers to finish the Guided Grad-CAM visualization and find that the final classifier after fusion can capture more features of the input image. It means that the $c$-th classifier can fuse the knowledge of $(c-1)$ classifiers by using CDM and hence improve the performance of the $c$-th classifier.

## 4.5 Combined with Improved Techniques

To further validate the effectiveness of CDM and GCDM, we conduct experiments by combining them with improved techniques. IMTA [19] is advanced two-stage improved techniques, which proposes the Gradient Equilibrium, and Forward-backward Knowledge Transfer (FKT) algorithms for improving training of *adaptive deep networks*. We name the model that uses only Gradient Equilibrium as GE, and both GE and FKT (whole two-stage training) as IMTA. The results of *budgeted batch prediction* on ImageNet100 shown in Figure 8(a) can be analyzed as follows: **(1)** MSDNet$^{E-IMTA}$ is better than MSDNet$^{E-GE}$, proving the two-stage training method further enhancing the performance of MSDNet$^{E-GE}$. **(2)** MSDNet$^{E-IMTA}$-CDM is better than MSDNet$^{E-IMTA}$ and MSDNet$^{E-GE}$-CDM is better than MSDNet$^{E-GE}$, demonstrating CDM is effective and can be combined with current techniques for better performance. **(3)** MSDNet$^{E-GE}$-GCDM outperforms both MSDNet$^{E-IMTA}$ and MSDNet$^{E-IMTA}$-CDM and obtains the best performance, indicating the combination of GCDM and GE algorithm can form a good single-stage method to replace existing two-stage training IMTA.

## 4.6 Loss Functions and Calculation Costs

**Discussions on different loss functions.** We use $\mathcal{L}_u = \mathcal{L}_{JS} + \sum_{c=1}^{C} \lambda_1 \mathcal{L}_{CE}^c(p^c, y)$ (cross-entropy loss) to replace the loss function (Eq. 13) and observe the accuracy changes. The results are

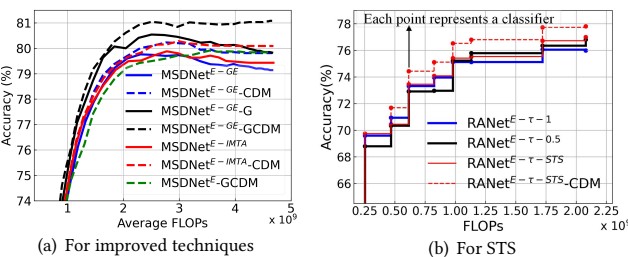

(a) For improved techniques          (b) For STS

**Figure 8: (a) The performance of combining CDM and GCDM with improved techniques. (b) The performance of using Stable Training Strategy (STS) in GCDM (values of $\tau_2$ or $\tau_1$ are 1 and 0.5).**

shown in Figure 9. The conclusion is our CDM isn't sensitive to the choice of loss function, as it consistently yields stable performance improvements when using other loss functions, e.g., cross-entropy loss. This further proves our ideas' effectiveness.

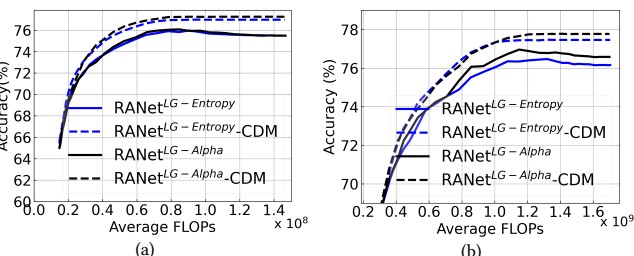

(a)          (b)

**Figure 9: Exploring the influence of loss function on our proposed method (*Budgeted batch prediction* on CIFAR100 and ImageNet100 datasets).**

**Discussions on Calculation Costs.** During the training stage, the additional cost is the added loss function Eq. 14, which can be negligible. During the inference stage, the extra computation cost comes from our uncertainty-aware fusion scheme between two adjacent classifiers. In fact, such extra computation costs also are negligible. Specifically, for inferring 50,000 images on the ImageNet1000 dataset, MSDNet with no fusion (*MSD*) requires **330.16 seconds**, while our method MSDNet-CDM with fusion (*MSD-CDM*) takes **330.37 seconds**. Similarly, RANet with no fusion (*RAN*) requires **342.15 seconds**, while our method RANet-CDM with fusion (*RAN-CDM*) takes **342.53 seconds**.

## 5 Conclusion

In this paper, we propose Collaborative Decision Making (CDM) and Guided Collaborative Decision Making (GCDM) to improve the classification performance of *adaptive deep networks*. CDM incorporates an uncertainty-aware fusion method to fuse decisions of different classifiers based on their *reliability* (uncertainty values). We also introduce a balance term to alleviate the fusion *saturation* and *unfairness* issues caused by the evidential deep learning framework, hence enhancing CDM's fusion quality. GCDM is designed to further improve CDM's performance through regularized training over earlier classifiers using the last classifier. Extensive experiments on CIFAR10, CIFAR100, ImageNet100 and ImageNet1000 show that our proposed CDM module and GCDM framework can consistently improve the performance of adaptive networks.

# 6  Acknowledgements

This work is supported by the Ministry of Science and Technology of China, the National Key Research and Development Program (No. 2021YFB3300503), and the National Natural Science Foundation of China (No. 62076072).

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
