# OpenReview forum: "Enhancing Adaptive Deep Networks for Image Classification via Uncertainty-aware Decision Fusion"
_acmmm.org/ACMMM/2024/Conference — MM2024 Poster_

### Official Review · Reviewer_Q8qz · 2024-05-09

**Rating:** 5
**Confidence:** 3

**Summary:**

This paper comploys multiple classiffer heads among different layers to improve decision making. this is an interesting work.

**Strengths:**

A Collaborative Decision Making idea is proposed to improve the performance of popular adaptive deep networks and experiments demonstrate that is is effective.

**Limitations:**

the word size in figure is too bold.

**Suitability:**

3

---

### Official Review · Reviewer_RJuU · 2024-05-23

**Rating:** 5
**Confidence:** 2

**Summary:**

The paper introduces a novel method, the Collaborative Decision Making (CDM) module, aimed at enhancing the inference performance of adaptive deep networks for image classification. This is achieved through an uncertainty-aware decision fusion technique that improves the accuracy of a selected classifier by leveraging the reliability (uncertainty values) from other classifiers in the network.

**Strengths:**

1.The introduction of the Collaborative Decision Making (CDM) module is a significant advancement in adaptive deep networks. It uniquely addresses the limitations of existing methods by utilizing earlier classifiers when they are more accurate, thus optimizing computational resources and improving performance.
2. Technical Correctness and Evaluation: The paper's methodology is technically sound, evidenced by thorough experimental evaluations on the ImageNet datasets. The results demonstrate an accuracy improvement of 0.4% to 2.8% under varying computational constraints, which substantiates the effectiveness of the proposed approach.
3. Theoretical Approach: The use of evidential deep learning (EDL) to quantify uncertainty and manage decision fusion is a robust theoretical addition. It not only enhances classifier reliability but also introduces a balance term to mitigate fusion saturation and unfairness, which are common in EDL applications.
4 .Clarity: The paper is well-organized and clearly written, making complex concepts accessible to readers. Figures and tables effectively illustrate the methodology and results, enhancing the paper's overall clarity and impact.

**Limitations:**

Lack of Comparative Analysis: While the paper provides a comparison with state-of-the-art models, it primarily focuses on improvements over traditional single-head classifiers. A broader comparison with other multi-head approaches or networks utilizing different types of decision fusion could strengthen the argument.

**Suitability:**

2

---

### Official Review · Reviewer_Ah8Y · 2024-05-23

**Rating:** 5
**Confidence:** 2

**Summary:**

This paper proposes methods to improve the classification performance of adaptive deep networks, namely Collaborative Decision Making (CDM) and Guided Collaborative Decision Making (GCDM). Experiments on ImageNet, CIFAR10 and CIFAR100 datasets show that CDM and GCDM can consistently improve the classification accuracy of state-of-the-art adaptive networks like MSDNet and RANet.

**Strengths:**

+ The paper is clearly written and easy to follow. The key ideas and contributions are described in detail.
+ The related work review is concise and accurate.
+ The authors publicize the training and evaluation code.
+ Consistent performance improvement can be observed from the experiments for different baseline models.
+ Sufficient ablation studies for different modules and factors.

**Limitations:**

- The illustration of CDM and GCDM in Figure 3 could be improved. They are not clear enough to show the idea.

**Suitability:**

2

---

### Official Review · Reviewer_k3iZ · 2024-05-24

**Rating:** 5
**Confidence:** 3

**Summary:**

This manuscript presents two methods, Collaborative Decision Making (CDM) and Guided Collaborative Decision Making (GCDM), to improve the performance of adaptive deep neural networks. CDM works by fusing the decisions of multiple classifiers within the network based on their uncertainty, using an uncertainty-aware attention mechanism to give more weight to more reliable classifiers. GCDM further enhances CDM by using regularized training, where the final classifier in the network guides the training of earlier classifiers for improving their accuracy without significantly impacting their diversity. The experiments are conducted on four datasets: CIFAR-10, CIFAR-100, ImageNet-100, and ImageNet-1000 and show that the proposed methods enhance the performance of the selection of adaptive deep networks involved in the experiments.

**Strengths:**

The concepts presented in the manuscript are comprehensible and straightforward. The writing quality is excellent. The paper appropriately references and utilizes relevant sources. The treatment of the subject is complete.

The approaches are orthogonal. Different from existing work in the literature, the method fuses the decisions of multiple classifiers using an uncertainty-aware approach and regularized training for optimizing the use of all classifiers' decision information.

The introduction of the balance term seems to have a positive impact on the fusion saturation and unfairness issues in CDM by slowing down the fusion process and improve the performance in subsequent fusion operations.

Ablation study is performed. The manuscript tested: 1) the impact of CDM and GCDM over the MSDNet literature approach, showing that both improve the baseline results, 2) the impact of the regularized training, showing that it doesn’t negatively influence the diversity of early classifiers and 3) the impact of the stable training strategy, showing that it can counteract the performance instability issues in training caused by either too long or too short distances between early classifiers and the final classifier.

The manuscript also experimented combining the proposed CDM and GCDM with Gradient Equilibrium and Forward-backward Knowledge Transfer literature approaches, showing their compatibility with current techniques for improved performance.

The source code is provided for reproducibility.

**Limitations:**

Missing insights into potential failure scenarios where the proposed methods may not perform optimally would enable a better understanding of the limitation of the approaches.

Minors
A few typographical errors exist in the manuscript, for example:
Row 919: “ ... their reliability(uncertainty values)” -> missing space between reliability and "(".
Row 836: missing before the reference;
etc.

**Suitability:**

2

---

### Meta-Review · Area_Chair_Yq9y · 2024-06-26

**Recommendation:** Accept (Poster)
**Confidence:** 5

**Metareview:**

Excellent work on uncertainty-aware decision fusion albeit that some minor improvements are still possible.